# *BRCA2* Pre-mRNA Differential 5′ Splicing: A Rescue of Functional Protein Properties from Pathogenic Gene Variants and a Lifeline for Fanconi Anemia D1 Patients

**DOI:** 10.3390/ijms26146694

**Published:** 2025-07-12

**Authors:** Roberto Paredes, Kiran Batta, Daniel H. Wiseman, Reham Gothbi, Vineet Dalal, Christine K. Schmidt, Reinhard Kalb, Stefan Meyer, Detlev Schindler

**Affiliations:** 1Epigenetics of Haematopoiesis Laboratory, Division of Cancer Sciences, School of Medical Sciences, Faculty of Biology, Medicine and Health, University of Manchester, Manchester M13 9PL, UK; roberto.paredes@manchester.ac.uk (R.P.); kiran.batta@manchester.ac.uk (K.B.); daniel.wiseman@manchester.ac.uk (D.H.W.); 2Manchester Cancer Research Centre (MCRC), Division of Cancer Sciences, School of Medical Sciences, Faculty of Biology, Medicine and Health, University of Manchester, Manchester M13 9PL, UK; christine.schmidt@manchester.ac.uk; 3Department of Paediatric Haematology and Oncology, Royal Manchester Children’s Hospital, Manchester M13 9WL, UK; 4Genome Stability Laboratory, Division of Cancer Sciences, School of Medical Sciences, Faculty of Biology, Medicine and Health, University of Manchester, Manchester M13 9PL, UK; 5Department of Human Genetics, Biozentrum, University of Würzburg, 97074 Würzburg, Germany; reinhard.kalb@uni-wuerzburg.de; 6Young Oncology Unit, The Christie NHS Foundation Trust, Manchester M20 4BX, UK

**Keywords:** BRCA2, splicing, Fanconi anemia, DNA repair

## Abstract

Fanconi anemia (FA) is a DNA repair deficiency disorder associated with genomic and chromosomal instability and a high cancer risk. In a small percentage of cases, FA is caused by biallelic pathogenic variants (PVs) in the *BRCA2/FANCD1* gene, defining the FA-D1 subtype. Experimental and epidemiologic data indicate that the complete absence of BRCA2 is incompatible with viability. Therefore, cells from individuals affected with FA caused by biallelic *BRCA2* PVs must have a residual BRCA2 function. This activity may be maintained through hypomorphic missense mutations, translation termination–reinitiation associated with a translational stop mutation, or other non-canonical or uncommon translation initiation and elongation events. In some cases, however, residual BRCA2 function is provided by alternatively or aberrantly spliced *BRCA2* transcripts. Here, we review and debate aspects of the contribution of splicing in the 5′ segment to BRCA2 functions in the context of PVs affecting this largely intrinsically disordered protein region, with a focus on recent findings in individuals with FA-D1. In this Perspective, we also discuss some of the broader biological implications and open questions that arise from considering 5′-terminal *BRCA2* splicing in light of old and new findings from FA-D1 patients and beyond.

## 1. Introduction

Fanconi anemia (FA) is a mostly recessive genetic disease with congenital developmental abnormalities, bone marrow failure, and an increased susceptibility to cancer. The disorder is caused by the germline disruption of any one of more than 20 *FANC* genes [1,2]. The *FANC* genes play a vital role in maintaining genomic and chromosomal integrity and stability. These genes also play a crucial part in the cellular response to DNA interstrand cross-links (ICLs), which are induced by intrinsic factors, such as oxidative or other free radical reactions and various aldehydes. Extrinsic factors, namely chemotherapeutics and carcinogens, including platinum compounds, mitomycin C (MMC), and diepoxybutane (DEB), likewise produce ICLs. These agents are also used in gold-standard laboratory assays for FA diagnosis.

The proteins encoded by *FANC* genes interact within the FA/BRCA pathway, which is part of the DNA damage response network. This pathway includes the upstream (pre-D2 or early) FA core complex proteins that facilitate the monoubiquitination of the FANCI and FANCD2 proteins, forming a distinct complex. This complex interacts with DNA and connects to the downstream (post-D2 or late) FA proteins. These proteins include PALB2/FANCN and BRCA2/FANCD1 [3], among others, and they link the FA pathway to homology-directed DNA repair [4].

## 2. Specific Roles and Issues of BRCA2 in Fanconi Anemia

FA caused by biallelic pathogenic variants (PVs) in *BRCA2/FANCD1* (OMIM *600185) accounts for only about 3% of FA cases [3,5], but it is clinically significant because it relates the underlying FA/BRCA pathway defect to hereditary breast and ovarian cancer (HBOC), due to monoallelic *BRCA2* PVs [6]. Some apparently sporadic cancers in the pediatric population may also be associated with monoallelic *BRCA2* PV carrier status [7,8]. Since its initial description in 2002 [9], FA caused by biallelic *BRCA2* variants has now been reported in over 90 individuals, as recently summarized in 2023 [10]. FA-D1, the corresponding subtype of FA, is typically associated with severe phenotypic manifestations of FA [11] and characteristically has a high incidence and a typical spectrum of blastoma-type embryonic cancers (‘embryoma’) and acute myeloid leukemia (AML). These findings suggest a critical role for intact BRCA2 in preventing malignant transformation during early development and hematopoiesis [10,11,12,13]. Complete gene knockout of the *BRCA2* gene is not viable in humans or mice [14,15], illustrating the essential function of BRCA2 in normal germ development and embryogenesis. Biallelic combinations of both *BRCA2/FANCD1* null alleles or other disruptive alleles are thought to result in prenatal lethality in many affected offspring of carrier parents, leading to reduced fertility and early pregnancy loss. The observed carrier frequency of *BRCA2* PVs indirectly supports this notion. It ranges from 0.42% to 0.69% in the general population [16,17], and is even higher in certain geographic and ethnic populations. Therefore, the expected incidence of FA-D1 patients with biallelic PVs in *BRCA2* is predicted to be 4 to 12 per million. However, this rate contrasts starkly with the documented prevalence of FA cases across all subtypes, which stands at four to seven per million live births [18], with a mere 3% of these cases being classified as FA-D1 [11]. These epidemiological data suggest that individuals with *BRCA2*-associated FA who survive to birth and beyond are severely underrepresented and must have some residual, essential BRCA2 function.

The human *BRCA2* gene (reference sequence NM_000059.4(BRCA2)) is located on the long arm of chromosome 13 (13q12.3). Consisting of 27 exons, the gene spans approximately 84 kb of DNA. The BRCA2 protein comprises 3418 amino acids (aa) and plays an essential role as a caretaker in maintaining genomic and chromosomal integrity, as well as cellular survival. It facilitates the repair of DNA double-strand lesions, including interstrand crosslinks (ICLs), through homologous recombination (HR), and also performs other critical cellular functions [19,20]. BRCA2 forms a homodimer that participates in the DNA damage response and indirectly promotes cell proliferation through complex and dynamic interactions with other proteins and DNA [21,22]. Figure 1A illustrates the partially characterized functional properties of the N-terminal region of the BRCA2 protein. The outermost N-terminal portion contains a 19 aa binding motif spanning residues 21 to 39, encoded by the 3′ end of *BRCA2* exon 2 and the 5′ portion of exon 3. This domain mediates interaction with the partner and localizer of BRCA2, PALB2, a protein that is essential for BRCA2 function [23,24]. The PALB2 antagonist, EMSY, which inhibits HR repair [25], also interacts with BRCA2 through this epitope, with its binding site mapped to residues 23 to 44 of BRCA2 [26,27]. Apart from these domains, the N-terminal region of the BRCA2 protein contains several large, highly unstructured stretches interspersed with short, transient α-helices [28] that are likely to adopt stable conformations only upon engagement with partner proteins. These intrinsically disordered regions enable BRCA2 to explore a wide range of conformational spaces and dynamic domain arrangements that serve as hubs for macromolecular complexes with distinct cellular functions (Figure 1B, left) [29,30]. This region is adjacent to another N-terminal DNA-binding domain, encoded by exon 10, that promotes RAD51-mediated HR [31]. Another functionally essential region of BRCA2, outside of the N-terminal region, is encoded by exon 11. This region contains the BRC repeats necessary for RAD51 binding and nucleation [20]. Lethality appears to occur when both alleles have PVs in the gene sites that are important for vital BRCA2 protein functions, such as the PALB2 binding site or the BRC repeats, and when there is no rescue mechanism. Such a mechanism, if present, may differ in the N-terminal region from that in other BRCA2 regions, and may vary depending on the PV’s exact site and type [32]. FA-D1 patients with biallelic PVs in exon 11 or other mandatory *BRCA2* regions have rarely been reported, despite the relatively uniform distribution of monoallelic PVs across the gene in hereditary breast and ovarian cancer (HBOC) patients [11,33,34,35]. These observations raise the key question of how FA-D1 patients with both alleles carrying pathogenic mutation 5′ to exon 11, upstream of the region encoding the vital BRC repeats, can be rescued, considering alternative or aberrant splicing.

## 3. *BRCA2* Variant 5′ Splicing Can Rescue DNA Repair Functions

The mechanism by which several FA-associated PVs in *BRCA2* confer viability was first investigated in a landmark study by Biswas et al. [36]. They examined cells from an FA patient with the *BRCA2* c.631+2T>G PV (previously c.864+2T>G; IVS7+2T>G), in a compound heterozygous status together with the c.3827delGT PV in exon 11. The authors identified an interesting aberrant splicing product that skipped the sequence related to exons 4 to 7 (*BRCA2*^Δ^^4–7^; Figure 1C). In addition to the expected aberrant splicing product that skips only the exon 7 sequence (*BRCA2*^Δ^^7^), resulting in a frameshift and premature protein truncation, the newly identified product was found alongside other differential splicing products from the *BRCA2* 5′ terminus. *BRCA2*^Δ^^4–7^ retained the reading frame and was shown to support RAD51 relocalization, nuclear repair foci formation, and other important functional properties of BRCA2 that are critical for viability. Notably, *BRCA2*^Δ^^4–7^ was expressed at higher levels in cells harboring the compound heterozygous c.631+2T>G/c.3827delGT PVs than in cells with the c.631+2T>G PV alone or in unrelated cells. Finally, the *BRCA2*^Δ^^4–7^ transcript was markedly reduced in the myeloid leukemia cell lines SB1685 and SB1690 from the FA patient with the c.631+2T>G/c.3827delGT PVs. Based on these observations, Biswas et al. [36] speculated that the FA patient in question survived due to the partial BRCA2 function provided by *BRCA2*^Δ^^4–7^. They also proposed that leukemia may have occurred due to a reduction in the *BRCA2*^Δ^^4–7^ transcript in certain hematopoietic cells. Subsequently, this variant was found to be necessary for tumor-free survival in *Brca*^Δex^^4–7^/*Brca*^Δex^^4–7^ knock-in mice [37].

Another study examined SBRes, a clone of the AML cell line SB1690CB [38], from the patient investigated by Biswas et al. [36]. Compared to SB1690CB, SBRes had acquired MMC resistance through prolonged exposure to low doses of MMC. In addition to MMC insensitivity, an in vitro analysis revealed that the functional properties of BRCA2 were restored, including the recovery of RAD51 foci formation and HR competence [39]. SBRes expresses the previously unknown *BRCA2* splice variant lacking the sequences corresponding to exons 5 and 7 (*BRCA2*^Δ^^5+7^; Figure 1C), as well as the *BRCA2*^ΔE^^5+7^-encoded protein.

In an intriguing and informative recent case study, Radulovic et al. describe another family affected by *BRCA2*-associated FA [40]. The report presents three affected siblings, who are the offspring of a consanguineous couple and are all homozygous for the PV c.469A>T, located 7 bp upstream of the 3′ end of *BRCA2* exon 5 (Figure 1 and Figure 2A).

The three siblings reported in this study survived beyond birth, although, as the authors point out, c.469A>T represents the most 5′-located homozygous *BRCA2* nonsense PV ever observed. Thus, the Radulovic et al. siblings broaden the scope of FA caused by homozygous or compound heterozygous *BRCA2* PVs in the region 5′ to exon 11. This finding confirms that even individuals affected by a very early homozygous nonsense PV in the 5′ region can survive if they have residual BRCA2 function, which is likely provided by some rescue mechanism that maintains viability. The predicted effect of the c.469A>T PV identified in this family is the premature translation termination, p.(Lys157*), resulting in the generation of a truncated, most likely nonfunctional, and unstable polypeptide, consisting of less than 160 aa from the N-terminal epitope of BRCA2 [40]. Radulovic et al. propose that the previously described aberrantly spliced *BRCA2* transcript, *BRCA2*^Δ4–7^ [40], which bypasses the nonsense PV, while maintaining the open reading frame in *BRCA2* mRNA [36,37], could be a possible rescue mechanism allowing survival. However, the expression of this transcript has not been demonstrated in cells from the affected siblings. Without mRNA data, important information regarding viable PVs in BRCA2, cellular rescue mechanisms, and essential domains is missing. Because this issue touches on several important topics, we sought experimental confirmation.

To identify *BRCA2* splice variants that could compensate for the c.469A>T nonsense mutation, we examined cDNA reverse-transcribed from the mRNA of blood-derived cells from patient 2 (individual III-5) in the study by Radulovic et al. [40] in a diagnostic setting (sample ID 14725), using the same approach as previously employed for the aforementioned FA-associated *BRCA2* splice variants [36,39]. We used primer pairs with sequences spanning the portion of the canonical *BRCA2* transcript BRCA-201 (ENST00000380152.8) that includes the sequence from exon 1/2 to exon 10, as specified in the legend to Figure 2B. We determined the transcripts expressed in a lymphoblast cell line from the homozygous c.469A>T individual III-5 [40]. For controls, we used cDNA from the AML cell line SB1690CB from the boy with the compound heterozygous germline PVs c.631+2T>G (previously c.864+2T>G; IVS7+2T>G) and c.3827delGT [36] and the subclone SBRes of this cell line [39]. Electrophoretic separation of the amplicons again showed multiple bands with different patterns in these cells. In SB1690CB and SBRs cells, we confirmed the splice variants *BRCA2*^Δ^^4–7^ and *BRCA2*^Δ^^5+7^, respectively (Figure 2B), as well as several other splice variants, as previously described [36,39]. Contrary to the hypothesis of the report by Radulovic et al. [40], the *BRCA2*^Δ4–7^ splice variant transcript was not detected in the cells harboring the homologous germline PV *BRCA2* c.469A>T. However, our analysis showed an additional *BRCA2* transcript that was slightly smaller than the full-length transcript and was absent from the controls (Figure 2B, red rectangle). Further analysis revealed the lack of the sequence associated with exons 4 and 5 (*BRCA2*^Δ^^4–5^), which also skipped the PV c.469A>T (Figure 2C). This aberrant splicing product retains the reading frame and predicts a BRCA2 protein missing 31 aa internally. Based on what is known about other *BRCA2* splice variants in this region, we suggest that this transcript contributes to the survival of the three affected siblings. Functional activity of BRCA2 has previously been demonstrated for the non-frameshift splice variant *BRCA2*^Δ4–7^ [36,37]. DNA repair competence has also been demonstrated earlier for the frame-preserving *BRCA2*^Δ^^5+7^ transcript. We have not yet studied the *BRCA2*^Δ^^4–5^ transcript comprehensively in mouse ES cells or knock-in mice. However, we propose that this non-frameshifting transcript from the same *BRCA2* region contributes to the residual BRCA2 function observed in sibling III-5, as reported by Radulovic et al. [40]. This is because exons 4 and 5 do not appear to be essential for certain BRCA2 functions that are maintained by the *BRCA2*^Δ4–7^ and *BRCA2*^Δ5+7^ transcripts, which also skip the sequence associated with exons 4 and 5.

The challenge of correctly assessing the effect of such splice variants and accurately interpreting their potential pathogenicity is exemplified by a recent publication by Nix et al. [41]: two *BRCA2* variants within consensus splice sites in the 5′ region, c.68-2A>G and c.425G>T, involving exons 3 and 4, respectively. Initially, complex interpretation supported a likely pathogenic classification by the ACMG guidelines. However, aberrant splicing was shown to produce at least one in-frame and potentially functional transcript in both cases. This led the authors to consider reclassification. Notably, c.68-2A>G was associated with the use of a cryptic splice acceptor six nucleotides into exon 3 for an in-frame transcript and c.425G>T, among others, with the in-frame transcript *BRCA2*^Δ^^4–5^, which was also observed to be related to the previously mentioned PV *BRCA2* c.469A>T.

## 4. *BRCA2* Variant 5′ Splicing—Buying Survival with Cancer Risk

Extensive alternative splicing naturally occurs in the 5′ region of *BRCA2*, as is well known [42]. Demonstrating an association between the *BRCA2*^Δ^^4–5^ transcript and the *BRCA2* PV c.469A>T further illustrates the many possibilities offered by differential splicing in the *BRCA2* 5′ region. We analyzed variant splicing using the SpliceVault web portal (https://splicevault.org/ accessed on 10 March 2025) and summarized the most common usage of cryptic splice sites in Table 1 to further support this concept. Conversely, Spliceator (http://www.lbgi.fr/spliceator/ accessed on 10 March 2025) proactively predicts splice site usage in cases of sequence variants. These tools use deep learning approaches and convolutional neural networks. Sequence variants affecting splicing elements could conceivably alter the balance of canonical, alternative, and aberrant splice products. However, this may be limited by the removal of some splice products via nonsense-mediated mRNA decay. We obtained a list of differential splice products associated with PVs in *BRCA2* exons 3 to 7 from our data and supplemented it with a literature search. This list is shown in Table 2. The ability of sequence variants in this region to affect the identity, number, pattern, or abundance of variant splice products is consistent with the idea that this region contains many splicing enhancers and silencers, as well as cryptic and canonical splice sites at the intron boundaries of the numerous small exons. All of these elements may contribute to splicing diversity. Aberrant splice products can have several possible effects. First, they can skip a pathogenic mutation. Second, they may produce a gapped transcript, resulting in the loss of genetic information from one or more small exons in this region while maintaining an intact reading frame. This allows the transcript to be translated into a BRCA2 protein with partially preserved functions. It may not be a coincidence that variant transcripts from this region can be translated into proteins with limited functions, since several exons are short, resulting in minimal loss of genetic information. Third, frameshifting and premature translation termination can produce unstable, truncated proteins that may still have regulatory functions. Nevertheless, as mentioned earlier, in rare instances, one or more differentially spliced transcripts may impart BRCA2 activity in the presence of pathogenic variants (PVs) in the *BRCA2* 5′ region. However, these transcripts may not prevent all of the clinical and cellular features of FA or protect cells against malignant transformation early in life.

The sequence commonly missing from variantly spliced isoforms of the *BRCA2* transcript in this region in association with FA is that of exon 5, which encodes a 17 aa sequence. The amino acid sequences encoded by exons 4 and 6 of *BRCA2* also appear to be of limited importance for survival, as their sequences are also skipped in the *BRCA2*^Δ^^4–7^ transcript. The most 5′ region represents the PALB2 binding domain [24] and overlaps with the sequence that has been implicated in the interaction with the EMSY repressor protein [26]. The extent to which PALB2 binding or EMSY interaction is altered in proteins translated from variant *BRCA2* transcripts, whose tertiary structure may affect PALB2 or EMSY affinity, has not been studied in detail. However, as PALB2 binding has been shown to be essential for suppressing malignant transformation. Subtle dysfunction could be biologically relevant [43] and very important given the high rate of embryonic cancers in BRCA2-associated FA.

Another function of BRCA2 associated with the N-terminal region is mediated by the aa residue T207, which is located in one of many short linear motifs (SLIMs) in BRCA2 (Figure 1A). T207 is one of several serine and threonine residues that are phosphorylated at the G2/M transition by the cyclin-dependent kinases CDK1/2 (including S70, T77, and S93) and the polo-like kinase PLK1 (including S193, S206, T207, T219, and T226) [44]. Although not all the functions of these phosphorylation events have yet been identified, they are highly conserved (Figure 1B). In particular, phosphorylation of the exon 7-encoded aa T207 promotes kinetochore microtubule attachment stability, facilitates chromosome alignment [44,45,46], and has been suggested to modulate cell cycle-dependent phosphorylation and activity of RAD51 [47]. The region containing T207 is not represented in several splice variants, including *BRCA2*^Δ^^4–7^ and BRCA2^Δ5+7^, which nevertheless confer some rescue of sensitivity to MMC and cross-linking agents [36,37,39]. It would, therefore, be interesting to systematically investigate which BRCA2 functions are affected and to what extent when T207 or exon 7 is lost. Another possibility is that some of the functions of BRCA2, such as interaction with RAD51, appear to depend on the homodimerization of the BRCA2 protein [22]. How BRCA2 dimerization is affected by splice variants, and the consequences for survival, cell proliferation, DNA repair, and chromosome segregation remain to be elucidated.

## 5. Recurrent Cancer Signatures but Variable Organ Manifestation for the Same PVs—A Paradox of *BRCA2* Variant 5′ Splicing

The data summarized here support the concept that certain variantly spliced *BRCA2* transcripts from the 5′ segment of *BRCA2* may confer BRCA2 activity for survival in the presence of PVs in the region spanning exons 3 to 7 or beyond, but do not prevent clinical manifestations of FA or malignant transformation. In this context, it is noteworthy that observations in large cohorts have shown that individual FA-associated *BRCA2* PVs may have an organ-specific cancer signature. For example, the germline PV c.631+2T>G (previously c.864+2T>G; IVS7+2T>G) has been strongly associated with AML in early childhood when present in homozygous or compound heterozygous status, whereas the PV c.658_659delGT (previously c.886delGT), also found in homozygous and compound heterozygous status, seems to be closely associated with brain tumors (unpublished observation), mostly medulloblastoma, and rarely with hematologic neoplasms in FA-D1 patients [10,11,48]. However, the three siblings in the family reported by Radulovic et al. individually developed distinct sets of a total of eight different neoplasms, including AML, nephroblastoma, hepatoblastoma, leukemia, and brain tumors, with little overlap between the siblings, despite their identical *BRCA2* mutation [40]. Radulovic et al. suggest that the occurrence of additional somatic events, maybe comparable to the *PTPN11* gain-of-function mutation c.1508G>T as a driver for juvenile myelomonocytic leukemia (JMML) in their patient 1, may have contributed to the development of specific embryonic tumors, but they did not further speculate on what these events might be specifically in *BRCA2*-associated malignancies of tissues or organs other than myeloblasts in JMML and how they might be caused.

In addition to splice variants, it is likely that modifier genes and genetic background, which have been shown to modulate and prolong embryonic survival in *Brca2*-null mice [49], as well as possibly chance events, contribute to the emerging cancer phenotype. To date, the exact contribution and role of splice variants, which may also vary in abundance and expression dynamics, have not been experimentally and systematically investigated and are, therefore, poorly understood. In this regard, it remains puzzling how *BRCA2* PVs located within the first few hundred nucleotides of the *BRCA2* coding sequence, which appear to generate functional “null” alleles, can be attenuated, presumably at least with a contribution of differential splicing, to preferentially cause certain cancers from a broad possible spectrum [16]. Extended data on *BRCA2* splicing in the presence of different PVs, to be collected by large-scale RNA sequencing, may confirm a previously suggested, potentially tissue-specific role of splice variants in cancer development, as discussed in the context of *BRCA2* PV carrier status for breast, ovarian, and pancreatic cancer [50,51]. The abundance and stoichiometry of splice-variant proteins may be another factor influencing the cancer phenotype in *BRCA2*-associated FA, which, so far, has not been investigated and would warrant further experimental interrogation.

## 6. Implications of *BRCA2* Differential 5′ Splicing in Non-FA Tissues

Variant transcripts from the *BRCA2* 5′ region have been detected in normal and *BRCA2*-PV-affected, blood-derived, and breast tissue-derived cells and cell lines [42]. As the term variant implies, they can modulate, edit, or alter the sequence information encoded at the N-terminal region of BRCA2. This occurs in an initially undirected manner, only to later prove to be unproductive or productive for certain cellular processes. However, only recently have variant *BRCA2* transcripts, selected from the ClinVar database of disease-associated human genetic diversity and the literature, been functionally analyzed. Several of them were first detected in *BRCA2*-disrupted FA-D1 patient-derived cells and may have functional relevance. They may attenuate the pathogenicity of *BRCA2* variants in FA-D1 patients, but also function in the presence of a *BRCA2* wild-type allele in *BRCA2*-PV carriers and may even have some effect in non-FA tissues. With the first demonstration of variant splicing of *BRCA2* in FA-D1 cells, associated with biallelic *BRCA2* PVs [36], an intriguing observation was the detection of variants that were found in cells from the affected boy with c.631+2T>G (previously c.864+2T>G; IVS7+2T>G), as well as in cells from a carrier of this PV and *BRCA2* wild-type cells [36]. The presence of these transcripts in *BRCA2* PV carriers is unlikely to represent an adaptive response in heterozygotes, since a single wild-type *BRCA2* allele provides sufficient functional BRCA2 activity to prevent clinical manifestations of FA and cellular DNA cross-linking sensitivity. Differential *BRCA2* splicing may also have normal physiological functions for a dynamic fine-tuning of BRCA2 functions in normal tissues. However, in the presence of PV-associated impaired BRCA2 functionality, differential splicing becomes a source of BRCA2 activity, and, with certain PV locations and combinations, might become the sole source of BRCA2 activity.

A role for *BRCA2* splice variants in *BRCA2* wild-type ovarian cancer has been suggested by a recent study investigating the effect of the splicing factor small nuclear ribonucleoprotein B (SNRPB) [52]. SNRPB, a core component of the spliceosome, has been identified as a critical driver of ovarian cancer. SNRPB was found to be upregulated in ovarian cancer cells, promoted tumor growth, supported canonical *BRCA2* splicing, and suppressed aberrant exon 3 skipping, thereby affecting the response to cisplatin. *SNRPB* knockdown inhibited ovarian cancer cell proliferation, downregulated DNA replication and HR genes, and promoted aberrant *BRCA2* exon 3 skipping, leading to a loss of the PALB2 binding domain and further impairment of HR. Thus, *SNRPB* silencing rescued cellular cisplatin sensitivity with the reappearance of a *BRCA2*^Δ^^3^ transcript [52] and decreased the levels of wild-type *BRCA2* mRNA. This study adds to previous observations regarding *BRCA2* splicing in ovarian cancer [53]. It provides further evidence that *BRCA2* splicing may be regulated by factors outside the *BRCA2* gene, and that a loss of coordinated splicing and *BRCA2* expression by *SNRPB* depletion suppresses ovarian cancer cell proliferation and rescues chemotherapy response. However, the opposite also seems possible, namely that, in cases of *BRCA2*-associated cancers with acquired chemotherapy resistance, PVs that do not allow for a rescue mechanism may confer drug insensitivity despite variant splicing [54].

Intrigued by these observations, we expanded our analysis, again using primer pairs spanning the entire sequence of exons 2 to 9 and focusing on cells and tissues with wild-type *BRCA2*. We compared the most extensively studied human breast adenocarcinoma cell line, MCF-7, with other human tissues. In MCF-7 cells, as previously reported [42], and also in human fetal brain and liver tissues (purchased from Agilent), which we examined because embryonic-type cancers are often associated with FA-D1, we again detected multiple transcripts in what are, by all accounts, tissues with the wild-type *BRCA2* genotype. All of these cell and tissue types showed variant transcripts from the 5′ segment of *BRCA2* (Figure 3A,B), which were identified by individual sequencing (Figure 3C,D). The patterns of additional *BRCA2* transcripts differ significantly in the number and size of individual bands, suggesting tissue-specific splicing, although some transcript species are common. However, an inter-individual variation of splicing might also be possible as, so far, the same tissues from different individuals and different tissues in the same individual have not been comprehensively investigated. Certainly, such results warrant further investigation of the stoichiometry and its dependencies. However, the cell- and tissue-type-specific differences in the pattern of *BRCA2*-variant transcripts on a wild-type *BRCA2* background are a strong argument for some form of regulation and, thus, for a physiological role in gene expression. By analogy, scRNA sequencing tells us that every cell in every tissue or organ has its own adaptive strategy [55,56]. *BRCA2* splice variants could have direct effects at the transcript level, play a role in the fine regulation of *BRCA2* translation, or have an indirect impact at the protein level, modulating BRCA2 homodimerization or interactions with other proteins or DNA [22].

Here, we only discuss splicing events in the 5′ segment of *BRCA2*. However, other regions have also been shown to undergo variant splicing. For example, splicing modulators in cohesin-mutated AML led to *BRCA2* exon 12 skipping and an associated reduction in BRCA2 levels, with a corresponding increase in chemosensitivity [57], revisiting the previously suggested redundancy of *BRCA* exon 12, as exon 12 has been shown to be non-essential for survival [58,59,60]. As a brief note, it should be mentioned that in addition to differential splicing, there is abundant evidence for hypomorphic missense variants of *BRCA2*, as identified, e.g., by Biswas et al. and Mesman et al. [36,60], which result in only partial loss of BRCA2 functions. In an extreme case, this is also true for a nonsense variant identified as a polymorphic stop codon [61]. Finally, translation reinitiation may result in the rescue of BRCA functions [62], as has been shown in patients with Nijmegen breakage syndrome and disrupting NBN PVs [63]. In Table 3, we summarize such sites with similarity to the Kozak sequence for potential translation restart in the BRCA2 region associated with exons 1 to 9.

The large size of the BRCA2 protein has made studies of the protein products resulting from translation of variant transcripts from the 5′ region of the gene challenging, as many spliced-out sequences do not alter the size of the BRCA2 protein enough to allow for easy electrophoretic separation and Western blot detection [59]. Accordingly, there is a paucity of data directly addressing the presence, abundance, and functional assessment of the protein products resulting from the variant splicing of *BRCA2*. However, the BRCA2^Δ5+7^ protein has been detected by a mass spectrometry analysis of splice variant-specific tryptic peptides from MMC-induced, MMC-resistant AML cells SBRes, derived from the FA patient-derived AML cell line SB1690CB [39], and supports the notion that variant transcripts from the 5′ segment of *BRCA2* are at least partially translated into proteins. Initial studies for their functional characterization have been performed using genetic engineering approaches and data obtained from survival, complementation, and other functional assays [60], and animal studies [37] are emerging but need to be refined and placed in a clinical and therapeutic context.

## 7. Summary, Conclusions, and Outlook

Here, we review some clinical and genetic aspects of variant splicing in the *BRCA2* 5′ region in the context of FA. Differential splicing appears to confer a survival benefit to some patients with biallelic pathogenic variants (PVs) in this gene region. We identified a splice variant from this region that skips exons 4 and 5 in an FA patient of subtype D1 who is homozygous for the *BRCA2* PV c.469A>T. Additionally, other variant transcripts from the *BRCA2* 5′ region also retain the reading frame, skip the PV, and are expressed at the protein level. Given these findings, we propose the hypothesis that the 5′ part of the *BRCA2* gene naturally yields many splicing variants that fine-tune the BRCA2 functional expression levels. This is particularly important in the case of PVs in this region because they can cause an upregulation of *BRCA2* function-rescuing transcripts. However, the basis for this remains unclear, as there is currently no known regulatory mechanism responsible for the regional relaxation of splicing fidelity. The idea that function-saving alternative or aberrant transcripts are differentially expressed could have broader implications for other large genes composed of multiple exons. This suggests that such genes might tolerate certain splicing mutations by producing differential transcripts from domain-poor regions, which retain partial function. More generally, we highlight critical BRCA2 functions that may be affected by variant splicing and review experimental and theoretical aspects of differential 5′ splicing within *BRCA2*-associated FA, in *BRCA2* PV carriers, and normal tissue. Our observations and those of other groups warrant further experimental work on the functional properties of *BRCA2* splice variants. Single-cell RNA-sequencing approaches will reveal *BRCA2*-associated variant transcriptomes resulting from differential splicing in a manner specific to mutation, cell, and cancer types. Building on the discovery of the BRCA2^Δ5+7^ [39] and *BRCA2*^Δ4–7^ proteins, we are developing mass spectrometry approaches for proteins resulting from variant splicing to study their interactions, in addition to their functions. Adequate expression of a full-functioning BRCA2 protein is critical for maintaining disease-free survival, so studying the factors that control it, including the splice variants, is a high priority.

## Figures and Tables

**Figure 1 ijms-26-06694-f001:**
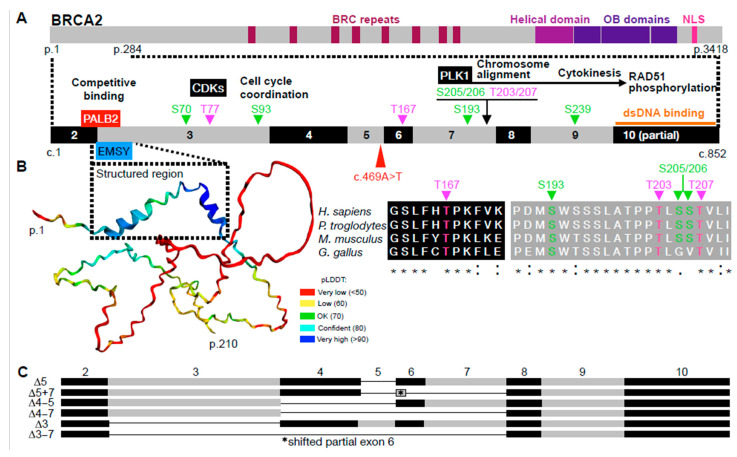
**The N-terminal region of BRCA2 (corresponding to exons 2 to 10)**. (**A**) Schematic representation of the BRCA2 protein, highlighting the N-terminal region (amino acids 1 to 284). Key features shown include wild-type phosphorylation sites, associated kinases, putative functional roles, and known interaction regions with PALB2 and EMSY. The location of the nonsense variant c.469A>T is marked with a red arrowhead. (**B**) (**Left**): AlphaFold-predicted structure of the N-terminal region of BRCA2 (amino acids 1 to 210), with confidence scores visualized by the predicted Local Distance Difference Test (pLDDT). (**Right**): Protein sequence alignment illustrating evolutionary conservation of phosphorylation sites within this region (using ClustalW). (**C**) Schematic of frequently observed, differently spliced *BRCA2* mRNA isoforms within the region of exons 2 to 10. Exon numbers are provided above.

**Figure 2 ijms-26-06694-f002:**
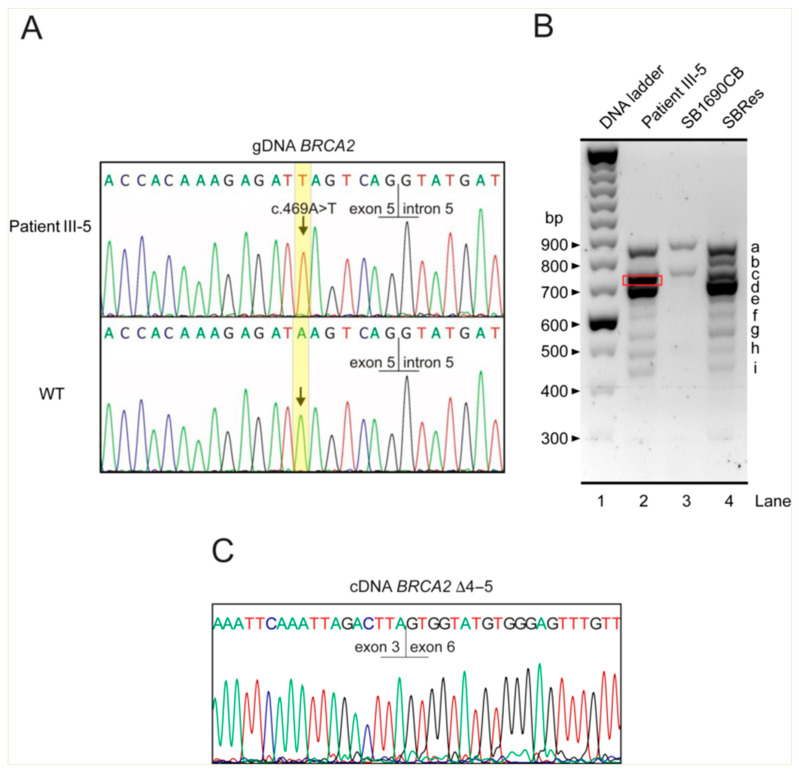
**Aberrant splicing caused by the *BRCA2* pathogenic variant (PV) c.469A>T,** as identified in cells from individual III-5 [40]. (**A**) Sanger sequencing electropherogram of PCR-amplified genomic DNA (gDNA) from individual III-5 confirms homozygosity for the *BRCA2* nonsense variant c.469A>T. (**B**) Electrophoretic separation of PCR-amplified products from cDNA using the primer pair 5′-TGGAGCGGACTTATTTACCAAG-3′ (forward, spanning exon 1/2 sequence) and 5′-TGCAGCTATTTACTTTAAATGAATTCCCTG-3′ (reverse, spanning exon 10 sequence). Lane 1: molecular weight ladder. Lane 2: LCL from individual III-5, showing a conspicuous band (indicated by a red rectangle). Lane 3: SB1690CB (FA-derived AML cells) carrying the germline splice donor PV c.631+2T>G (formerly c.864+2T>G; IVS7+2T>G) and the small frameshift deletion c.3827delGT in compound heterozygous status [36]. Lane 4: SBRes cells [38], displaying multiple, differentially spliced *BRCA2* isoforms, including (a) full-length (885 bp); (b) Δexon 5 (835 bp); (c) Δexon 7 (770 bp); (d) Δexons 5 + 7 (720 bp); (e) Δexons 5–7 with insTG (681 bp); (f) Δexon 3 (636 bp); (g) Δexons 4–7 (570 bp); (h) Δexons 3 + 7 (521 bp); and (i) Δexons 3 + 5 + 7 (471 bp). (**C**) Sanger sequencing of the conspicuous band from lane 2 (red rectangle) reveals an atypical splicing product in which the exon 3 sequence joins directly to the exon 6 sequence, indicating exon 4 and 5 skipping.

**Figure 3 ijms-26-06694-f003:**
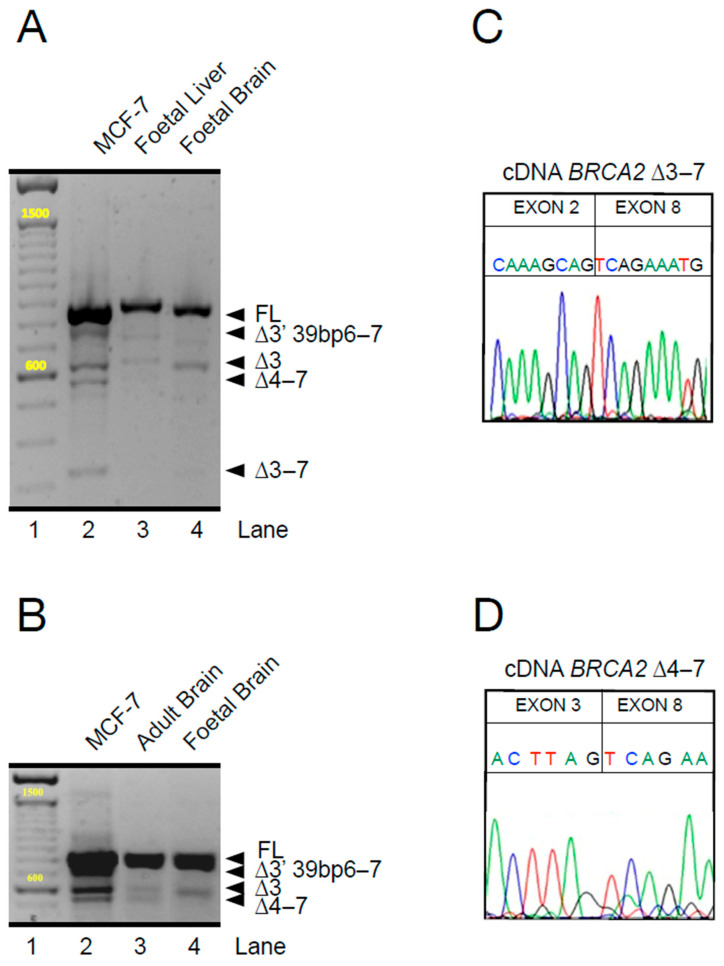
**Age- and cell-type-dependent differential splicing in the 5′ region of *BRCA2* in non-FA tissue.** (**A**,**B**) Separation of PCR-amplified cDNA spanning the transcribed region between exons 1/2 to 10. Represented are transcripts from MCF-7 cells, as well as transcripts from human fetal liver and brain in (**A**), and from adult and fetal brain tissue in (**B**) for comparison. Note the variable presence and abundance of several bands indicating splice variants. Size in bp of the major bands are indicated in yellow. (**C**,**D**) cDNA sequences of the variant transcripts *BRCA2*^Δ^^3–7^ and *BRCA2*^Δ^^4–7^.

**Table 1 ijms-26-06694-t001:** Use of cryptic splice sites in the region of *BRCA2* intron 3 to 7 among 335,663 publicly available RNA-seq samples.

Cryptic Splice Site (Position)	Exon/Intron (IVS) no.	Predicted Function	Sample Count (Frequency) ^1^
c.67+1254	IVS 2	acceptor	3467 (3.8%)
c.317-419	IVS 3	acceptor	120 (0.1%)
c.320	4	acceptor	7866 (7.7%)
c.425+381	IVS 4	acceptor	433 (0.4%)
c.426-208	IVS 4	donor	781 (0.7%)
c.426-24	IVS 4	acceptor	145 (0.1%)
c.632-187	IVS 7	donor	310 (0.3%)

^1^ Total number of variant transcript samples (or frequency), assessed exon-by-exon using the SpliceVault portal (https://splicevault.org/ accessed on 10 March 2025), which draws on entries from a large population database; threshold for display, variant number ≥ 100.

**Table 2 ijms-26-06694-t002:** Some variant *BRCA2* transcripts skipping sequences associated with exons 3 to 7 due to pathogenic sequence variants.

Exon no.	Splice Variant	Length (nt)	Reduction of Transcript Length (nt)	Effect on the Reading Frame
3	canonical	249	−	−
−	∆3	−	−249	in frame
	∆3–4		−358	out of frame
−	∆3 + 7	−	−364	out of frame
−	∆3 + 5 + 7	−	−414	in frame
−	∆3–7	−	−564	in frame
4	canonical	109	−	−
	∆4		−109	out of frame
−	∆4–5		−159	in frame
−	∆4–7		−315	in frame
5	canonical	50	−	−
−	∆5	−	−50	out of frame
−	∆5 + 7	−	−165	in frame
−	∆5–7, insTG	−	−204	in frame
6	canonical	41	−	−
7	canonical	115	−	−
−	∆7	−	−115	out of frame
8	canonical	50	−	−

Footnotes. ∆ indicates sequence skipping associated with the indicated exon(s). Highlighted in yellow: identified in FA-D1 patient-derived cells. Other variants were obtained from the literature searches.

**Table 3 ijms-26-06694-t003:** Exemplary translation initiation sites (TISs) of *BRCA2* exons 1 through 9 with a Kozak similarity score of 0.5 or higher ^1^.

TIS (CDS Position)	Exon No.	Kozak Similarity Score	Reading Frame Phase	Translation Option
c.1 (canonical start codon)	2	0.66	+1	yes
c.323	4	0.50	+2	no, in-frame TGA at c.359 (exon 4)
c.370	4	0.71	+1	yes
c.441	5	0.62	+3	no, in-frame TAA at c.468 (exon 5)
c.574	7	0.57	+1	yes
c.638	8	0.62	+2	unlikely ^2^, TGA at c.686 (exon 9)
c.707	9	0.7	+2	no, TGA at c.719 (exon 9)
c.710	9	0.67	+2	no, TGA at c.719 (exon 9)
c.728	9	0.62	+2	no, TGA at c.749 (exon 9)
c.791	9	0.74	+2	unlikely ^2^, TAA at c.827 (exon 10)

^1^ In silico analysis of RefSeq NM_000059.4 *BRCA2* using a Kozak similarity score algorithm for TISs with flanking nucleotides that closely match the Kozak consensus motif (https://www.tispredictor.com/tis#, accessed 10 March 2025) within the CDS corresponding to the first 9 exons. ^2^ Only if the normal reading frame is restored by aberrant splicing.

## Data Availability

Materials and further details of methods used in this study is available on request.

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
