# Peer review of "BRCA2 Pre-mRNA Differential 5′ Splicing: A Rescue of Functional Protein Properties from Pathogenic Gene Variants and a Lifeline for Fanconi Anemia D1 Patients"

_ijms, 2025, doi:10.3390/ijms26146694_

Round 1

Reviewer 1 Report

Comments and Suggestions for Authors

The authors identified a novel splice variant that may account for the phenotype of the patients described in previously published work. Based on this, they propose an interesting hypothesis: that the 5' part of the BRCA2 gene naturally yields many splicing variants to fine-tune its functional expression levels. This idea could have broader implications for other large genes composed of multiple exons, suggesting that such genes might tolerate certain splicing mutations by producing alternative transcripts that retain partial function.

This hypothesis is intriguing, especially because there is currently no known mechanism responsible for regional relaxation of splicing fidelity.

However, the manuscript would benefit from several revisions to improve clarity and accessibility to a broad readership.

Major comment:

The manuscript begins with a specific example highlighting an inconsistency in previous reports and introduces the possible role of a splice variant in resolving that inconsistency. This part is clearly presented, and the hypothesis drawn from the result is both logical and interesting.

However, the latter part of the manuscript (sections 4 and 5) relies on weaker experimental evidence and does not add substantial impact beyond the initial finding. The detailed description of the relationship between PVs and cancer is interesting. However, there is currently no experimental data demonstrating the functionality of protein products of these splice variants in cancers associated with PVs. As a result, the conclusions drawn in the latter half overlap with those already presented in the first half.

I suggest condensing the second half of the manuscript. It should be reframed to focus on discussing the expected consequences of relaxed splicing fidelity in the 5' region of BRCA2, rather than speculating on functional outcomes without supporting evidence. This approach would strengthen the manuscript by emphasizing the conceptual novelty without overstating the implications.

Minor points:

1Figure 1 legend: The legend states, “Arrow: position of the pathogenic variant c.469A>T,” referring to the bottom panel. However, the Arrow (actually, an arrowhead) appears in the top panel. Please revise the legend for clarity and also explain the meaning of the label "III-5/CNEN" on the arrowhead.

2Lines 138–139: The authors describe the previous study as "misleadingly referred to as 'early in exon 5.'" It may not be necessary to include this critique unless it is essential to the argument. Consider whether this comment contributes to the scientific point or could be omitted.

3Lines 343–348: This sentence is difficult to follow due to its complexity. Please consider rephrasing it for improved readability.

Author Response

Reviewer 1

The authors identified a novel splice variant that may account for the phenotype of the patients described in previously published work. Based on this, they propose an interesting hypothesis: that the 5' part of the BRCA2 gene naturally yields many splicing variants to fine-tune its functional expression levels. This idea could have broader implications for other large genes composed of multiple exons, suggesting that such genes might tolerate certain splicing mutations by producing alternative transcripts that retain partial function.

This hypothesis is intriguing, especially because there is currently no known mechanism responsible for regional relaxation of splicing fidelity.

We thank this Reviewer for his overall positive assessment of the manuscript.

However, the manuscript would benefit from several revisions to improve clarity and accessibility to a broad readership.

Major comment:

The manuscript begins with a specific example highlighting an inconsistency in previous reports and introduces the possible role of a splice variant in resolving that inconsistency. This part is clearly presented, and the hypothesis drawn from the result is both logical and interesting.

However, the latter part of the manuscript (sections 4 and 5) relies on weaker experimental evidence and does not add substantial impact beyond the initial finding. The detailed description of the relationship between PVs and cancer is interesting. However, there is currently no experimental data demonstrating the functionality of protein products of these splice variants in cancers associated with PVs. As a result, the conclusions drawn in the latter half overlap with those already presented in the first half.

I suggest condensing the second half of the manuscript. It should be reframed to focus on discussing the expected consequences of relaxed splicing fidelity in the 5' region of BRCA2, rather than speculating on functional outcomes without supporting evidence. This approach would strengthen the manuscript by emphasizing the conceptual novelty without overstating the implications.

We agree with the Reviewer that the manuscript in sections 4 and 5 summarises mainly clinical and experimental observations. However, the aspects of BRCA2 splicing that are addressed in these sections as intriguing and potentially biologically important, but we would consider them not recognised to a level these observations deserve. Our view appears to be shared also by Reviewer 2. These sections address separate aspects in malignant and normal tissue, in the context of FA and in sporadic cancers, and we would argue that they should be kept separately. We do emphasize the absence of functional data for these sections and possible explanations (e.g. To date, the exact contribution and role of splice variants, which may also vary in abundance and expression dynamics, have not been experimentally and systematically investigated and are therefore poorly understood. In this regard, it remains puzzling….). To address the Reviewer's point, we have changed the last sentence of section 4 as follows: The abundance and stoichiometry of splice variant proteins may be another factor influencing the cancer phenotype in BRCA2-associated FA, which has not been investigated and would warrant further experimental interrogation. In section 5 we review available experimental published evidence in malignant and normal tissue from other groups and our own work, and we feel that the descriptive narrative does not overstate the implications. We therefore would like to keep the two sections separate, and we feel that topics addressed in these sections warrant the descriptive review.

Minor points:

1 Figure 1 legend: The legend states, “Arrow: position of the pathogenic variant c.469A>T,” referring to the bottom panel. However, the Arrow (actually, an arrowhead) appears in the top panel. Please revise the legend for clarity and also explain the meaning of the label "III-5/CNEN" on the arrowhead.

We thank the Reviewer for pointing out these inaccuracies. We have edited Figure 1 accordingly and indicated the position of the variant c.469A>T and removed “III-5/CNEN”. We have also made some more changes to the figure to increase clarity, and indicated start and end amino acid of the AlphaFold figure

2Lines 138–139: The authors describe the previous study as "misleadingly referred to as 'early in exon 5.'" It may not be necessary to include this critique unless it is essential to the argument. Consider whether this comment contributes to the scientific point or could be omitted.

We agree with the Reviewer and have rephrased and removed the second half of the sentence (line 167 in the Revised version).

3Lines 343–348: This sentence is difficult to follow due to its complexity. Please consider rephrasing it for improved readability.

We agree with the Reviewer and have rephrased this section as follows Lines : Differential BRCA2 splicing may also have normal physiological functions for a dynamic fine-tuning of BRCA2 functions in normal tissues. However, in the presence of PV-associated impaired BRCA2 functionality differential splicing becomes a source of BRCA2 activity, which with certain PV locations and combinations might become possibly the sole source of BRCA2 activity. (lines 360-363 in the revised manuscript).

Reviewer 2 Report

Comments and Suggestions for Authors

Here, Paredes et al. (ijms-3677833) provide a detailed review of BRCA2/FANCD1 as a Fanconi anemia (FA) gene and of the role of splicing at the 5' end of BRCA2 in the rescue of pathogenic variants in a largely intrinsically disordered region of encoded proteins. Overall, this is an interesting and generally well-written review that highlights an under explored area related to FA-D1 patients and includes a brief look at tools to analyze variant splicing and an interesting section "4." on the association of particular pathogenic variants with specific cancer signatures. The manuscript concludes with open questions and future directions. Nevertheless, there are a couple of issues, as well as minor points, that should be addressed prior to potential publication.

Major points:

  1. Based on the diagram at the bottom of Figure 1, the nomenclature for specific splice variants is unclear. Labels to specific exons in this section of the figure and an examination of whether the names that are utilized match the diagram would make the review more accessible for readers that are not experts in splicing. In certain cases, it might help to explain what a particular term describes (for example is "4+5" different from "4-5"; if not, can consistent terminology be used since "3-7" is utilized elsewhere for skipping of exons 3-7?). Another example of a lack of consistency is that Figure 1 lists "5+7" and "4-7", respectively, while lines 166 and 168 list "5_7" and "4_7". Also, does transcript "3+5+7" in lines 169-170 contain exons 4 and 6? Please provide a clearer description for this. Further, lines 190-191 describe a sequence associated with the absence of "exons 4 and 5", but in line 199 this is described as del exons "4+5"; is this the same as del exons "4-5"? Also, in Figure 3A and 3B, what do "E6 + E7" indicate?
  2. Lines 367-378 on p. 10 describe results presented in Figure 3 and, on this basis, the authors conclude that the results provide evidence of "tissue-specific splicing" (line 376). This appears to be problematic since the differences in splicing patterns in MCF-7 cells vs human liver and/or human brain could instead be due to the particular cells/tissues having come from different individuals, especially since replicates from different individuals were not included.

Minor points:

  1. In lines 38-39, there is font usage that does not match the remainder of the text. 
  2. There are some minor issues with the presentation of Figure 1 that should be addressed. First, the order of the EMSY and PALB2 binding regions of the BRCA1 protein in Figure 1 do not match the description in the text (lines 82-87). Also, "shifted partial exon 6" in blue is too small to be readily legible in the lower part of Figure 1. Additionally, the last sentence of the legend ("Asterisks indicate elsewhere or previously characterized transcripts", lines 111-112) is unclear.  
  3. In line 142, "23" after Radulovic et al. is confusing since ref. 23 is Oliver et al.
  4. If all of the splice variants listed in Table 2 are pathogenic, the table might be clearer by adding "pathogenic" before "variants" in the title.
  5. Line 255 "derived from other FA-D1 patient" is unclear.  How about "derived from another FA-D1 patient" or "derived from other FA-D1 patients"?
  6. The statement, "By analogy, scRNA sequencing tells us that every cell in every tissue or organ has its own adaptive strategy" in lines 381-383 would be strengthened by addition of a reference(s).
  7. The statement "there is abundant evidence for hypomorphic missense variants of BRCA2, as identified e.g. by Biswas et al. and Mesman et al. [35,56], which rescue BRCA2 functions (lines 398-400) is unclear. Does a hypomorphic missense variant actually "rescue" BRCA2 function as opposed to merely displaying a "partial" loss-of-function? This may need rewording.
  8. The presentation might be clearer by the introduction of a new paragraph in line 405 beginning with "The large size of the BRCA2 protein ...". 
Comments on the Quality of English Language

There is a small number of places where the English utilization could and/or should be improved, as described in minor points above in comments to the authors.

Author Response

Reviewer 2

Here, Paredes et al. (ijms-3677833) provide a detailed review of BRCA2/FANCD1 as a Fanconi anemia (FA) gene and of the role of splicing at the 5' end of BRCA2 in the rescue of pathogenic variants in a largely intrinsically disordered region of encoded proteins. Overall, this is an interesting and generally well-written review that highlights an under explored area related to FA-D1 patients and includes a brief look at tools to analyze variant splicing and an interesting section "4." on the association of particular pathogenic variants with specific cancer signatures. The manuscript concludes with open questions and future directions.

We thank this Reviewer for his overall positive assessment of the manuscript.

Nevertheless, there are a couple of issues, as well as minor points, that should be addressed prior to potential publication.

Major points:

  1. Based on the diagram at the bottom of Figure 1, the nomenclature for specific splice variants is unclear. Labels to specific exons in this section of the figure and an examination of whether the names that are utilized match the diagram would make the review more accessible for readers that are not experts in splicing. In certain cases, it might help to explain what a particular term describes (for example is "4+5" different from "4-5"; if not, can consistent terminology be used since "3-7" is utilized elsewhere for skipping of exons 3-7?). Another example of a lack of consistency is that Figure 1 lists "5+7" and "4-7", respectively, while lines 166 and 168 list "5_7" and "4_7". Also, does transcript "3+5+7" in lines 169-170 contain exons 4 and 6? Please provide a clearer description for this. Further, lines 190-191 describe a sequence associated with the absence of "exons 4 and 5", but in line 199 this is described as del exons "4+5"; is this the same as del exons "4-5"? Also, in Figure 3A and 3B, what do "E6 + E7" indicate?

We thank the Reviewer for picking up inconsistencies in the used nomenclature, which was an oversight on our part. We have corrected the figures, the figure legends and the text and use “+” between exons if the deleted exons are not in continuation and “-“, if they are. We apologise for the confusion caused and tried to be consistent throughout the text and the figure legends.

  1. Lines 367-378 on p. 10 describe results presented in Figure 3 and, on this basis, the authors conclude that the results provide evidence of "tissue-specific splicing" (line 376). This appears to be problematic since the differences in splicing patterns in MCF-7 cells vs human liver and/or human brain could instead be due to the particular cells/tissues having come from different individuals, especially since replicates from different individuals were not included.

This is a valid point. We have changed the narrative as follows: The patterns of additional BRCA2 transcripts differ significantly in the number and size of individual bands, suggesting tissue-specific splicing, although some transcript species are common. However, an inter-individual variation might also be possible, as so far, the same tissue from different individuals and different tissue in the same individual has not been comprehensively investigated. (Lines 390 -393 in the revised manuscript)  

 Minor points:

  1. In lines 38-39, there is font usage that does not match the remainder of the text. 

 Corrected

  1. There are some minor issues with the presentation of Figure 1 that should be addressed. First, the order of the EMSY and PALB2 binding regions of the BRCA1 protein in Figure 1 do not match the description in the text (lines 82-87). Also, "shifted partial exon 6" in blue is too small to be readily legible in the lower part of Figure 1. Additionally, the last sentence of the legend ("Asterisks indicate elsewhere or previously characterized transcripts", lines 111-112) is unclear.  

We thank the Reviewer for picking up these inaccuracies. We have changed the position of PALB2 and EMSY, changed the colour and labelling of “shifted partial exon 6”. We removed the asterisks. In addition, we have introduced for figure 1 A, B and C, as this appears clearer than the “upper, middle and lower panels”. We have also added further detail with respect to amino acid number in the modelling illustration, and added exon numbers in C.

  1. In line 142, "23" after Radulovic et al. is confusing since ref. 23 is Oliver et al.

We thank the Reviewer for picking this up. We have removed the reference 23 at this point.

  1. If all of the splice variants listed in Table 2 are pathogenic, the table might be clearer by adding "pathogenic" before "variants" in the title.

We added “pathogenic “to the title.

  1. Line 255 "derived from other FA-D1 patient" is unclear.  How about "derived from another FA-D1 patient" or "derived from other FA-D1 patients"?

We reviewed the footnote and the highlighting and have changed the footnote for clarity as follows:

 Footnotes

∆ indicates sequence skipping associated with the indicated exon(s).

Highlighted in yellow: identified in FA-D1 patient derived cells. Other variant obtained from literature searches

  1. The statement, "By analogy, scRNA sequencing tells us that every cell in every tissue or organ has its own adaptive strategy" in lines 381-383 would be strengthened by addition of a reference(s).

We added the references: Hadadi et al and Sutzova et al.

  1. The statement "there is abundant evidence for hypomorphic missense variants of BRCA2, as identified e.g. by Biswas et al. and Mesman et al. [35,56], which rescue BRCA2 functions (lines 398-400) is unclear. Does a hypomorphic missense variant actually "rescue" BRCA2 function as opposed to merely displaying a "partial" loss-of-function? This may need rewording.

We agree with this point and have rephrased as follows:” …. which result in only partial loss of BRCA2 functions.” (lines 409-410 in the revised manuscript).

  1. The presentation might be clearer by the introduction of a new paragraph in line 405 beginning with "The large size of the BRCA2 protein ...". 

We agree and have inserted a paragraph.

On detailed review, guided by the Reviewers’ suggestions, we also have edited the subheadings for section 1, and 2 and 3. We renamed section two “Specific roles and issues of BRCA2 in Fanconi anemia”, and section 3 BRCA2 variant 5' splicing can rescue DNA repair functions.  We feel that these subheadings are a better choice for the sections, and hope that this is acceptable. We also removed the adjective “novel” for the BRCA2Δ4-5; because - while this transcript has not been detected associated in FA-derived tissue (new in that sense), - it has been described in another context, as we point out by discussing the work from Nix et al. at the end of section 2. Throughout the manuscript we made small changes to the text to improve the clarity and the English language. With respect to primers, we have added that the sequences would be available on request. We also would like to add additional funding which we by mistake omitted in the initial submission.

Round 2

Reviewer 1 Report

Comments and Suggestions for Authors

The authors have addressed my comments appropriately.

Author Response

We thank the reviewer for his constructive comments.

Reviewer 2 Report

Comments and Suggestions for Authors

The manuscript has been markedly improved by employing a more consistent nomenclature for particular splice variants, via revision of Figure 1, by adding certain references, and by considering "inter-individual variation in splicing" as a possible basis for the results obtained in Figure 3. Also, the expanded discussion of splice variants in section 7 (Summary, Conclusions and Outlook) is an interesting addition.

It should be noted that Fig. 3A in the revised version appears to be missing much of the information that was present in the original. This issue needs to be addressed.

There are several minor points that also need to be addressed:

  1. The font for "oxidative or other free radical ... chemotherapeutics and carcinogens" in lines 46-47 of the revised manuscript does not match that utilized elsewhere.
  2. For the sentence "Since its initial description..." (lines 75-78 of the revised manuscript), it appears that ref. 9 was published in 2002 (not "2003" as stated on line 76). 
  3. Please verify that when changes are accepted, that ", and is even higher ... residual, essential BRCA2 function." in lines 102-108 of the revised manuscript is part of the paragraph from 66-101.
  4. The appearance of "issue/issues" twice in the sentence beginning "Because this issue" (lines 222-223) makes it unclear.
  5. Would "red rectangle" be clearer than "red oblong", which occurs twice in the legend to Figure 2?
  6. There is a gap/unused space over a large part of p. 12.

Author Response

Response to Reviewer

Reviewer 2:The manuscript has been markedly improved by employing a more consistent nomenclature for particular splice variants, via revision of Figure 1, by adding certain references, and by considering "inter-individual variation in splicing" as a possible basis for the results obtained in Figure 3. Also, the expanded discussion of splice variants in section 7 (Summary, Conclusions and Outlook) is an interesting addition.

Response: We thank Reviewer 2 for the appreciation and kind words.

Reviewer 2: It should be noted that Fig. 3A in the revised version appears to be missing much of the information that was present in the original. This issue needs to be addressed.

Response: Thank you. This was a transmission error and was unintentional. We have replaced the damaged version with the original.

Reviewer 2: There are several minor points that also need to be addressed:

  1. The font for "oxidative or other free radical ... chemotherapeutics and carcinogens" in lines 46-47 of the revised manuscript does not match that utilized elsewhere.

Response: This seemed to be a matter of print editing by the publisher. We have changed the font of this sentence to match the rest of the text, Palatino Linotype.

Reviewer 2:

  1. For the sentence "Since its initial description..." (lines 75-78 of the revised manuscript), it appears that ref. 9 was published in 2002 (not "2003" as stated on line 76). 

Response: Of course, you are right. We have fixed it so that it now says "2002".

Reviewer 2:

  1. Please verify that when changes are accepted, that ", and is even higher ... residual, essential BRCA2 function." in lines 102-108 of the revised manuscript is part of the paragraph from 66-101.

Response: This section, "2. Specific Roles and Issues of BRCA2 in Fanconi Anemia" (lines 54–114 of the second revision), contains two paragraphs. The indicated text is at the end of the first paragraph (lines 72–79 of the second revision).

Reviewer 2:

  1. The appearance of "issue/issues" twice in the sentence beginning "Because this issue" (lines 222-223) makes it unclear.

Response: That’s true. We changed it to read as follows: "Because this issue touches on several important topics, we sought experimental confirmation (lines 176–177 of the second revision)."

Reviewer 2:

  1. Would "red rectangle" be clearer than "red oblong", which occurs twice in the legend to Figure 2?

Response: We agree and have changed "red oblong" to "red rectangle" in the text (line 196 of the second revision) and twice in the legend for Figure 2, B and C.

Reviewer 2:

  1. There is a gap/unused space over a large part of p. 12.

Response: Once again, it appears that there is a problem with the publisher's print editing. Hopefully, it will be fixed in the final version.

Comment: In addition to the suggestions from Reviewer 2, we have:

  1. Edited the text again to improve clarity and correct typos;
  2. Changed "FA genes" to "FANC genes" throughout the text;
  3. Standardized the figure number designations by omitting spaces, e.g., 1A, 2B, etc;
  4. Moved up the first paragraph of section 2 of the former version to become the second paragraph of the introduction (section 1) of the second revision;
  5. Included the primer sequences used for PCR amplification and detection of the BRCA2Δ4-5 transcript in the Figure 1 legend.

Please also kindly accept the changes with respect to funding that were omitted in the original submission and already included in the first revision.